# Neuroendocrine Tumors: A Comprehensive Review on Nutritional Approaches

**DOI:** 10.3390/cancers14184402

**Published:** 2022-09-10

**Authors:** Giovanna Muscogiuri, Luigi Barrea, Maria Celeste Cantone, Valentina Guarnotta, Rossella Mazzilli, Ludovica Verde, Claudia Vetrani, Annamaria Colao, Antongiulio Faggiano

**Affiliations:** 1Centro Italiano per la cura e il Benessere del Paziente con Obesità (C.I.B.O), Unità di Endocrinologia, Dipartimento di Medicina Clinica e Chirurgia, Università Federico II, 80131 Naples, Italy; 2Unità di Endocrinologia, Diabetologia ed Andrologia, Dipartimento di Medicina Clinica e Chirurgia, Università Federico II, 80131 Naples, Italy; 3Cattedra Unesco “Educazione alla Salute e allo Sviluppo Sostenibile”, Università Federico II, 80131 Naples, Italy; 4Dipartimento di Scienze Umanistiche, Università Telematica Pegaso, Via Porzio, Centro Direzionale, Isola F2, 80143 Naples, Italy; 5Department of Medical Biotechnology and Translational Medicine (BIOMETRA), University of Milan, 20157 Milan, Italy; 6Dipartimento di Promozione della Salute, Materno-Infantile, Medicina Interna e Specialistica di Eccellenza “G. D’Alessandro” (PROMISE), Sezione di Malattie Endocrine, del Ricambio e della Nutrizione, Università di Palermo, 90127 Palermo, Italy; 7Department of Clinical and Molecular Medicine, Sant’Andrea Hospital, Sapienza University of Rome, 00185 Rome, Italy

**Keywords:** neuroendocrine neoplasms, neuroendocrine tumors, cancer, diet, Mediterranean diet, ketogenic diet, fasting

## Abstract

**Simple Summary:**

Neuroendocrine neoplasms and their main subtype neuroendocrine tumors have an increasing incidence worldwide, associated with a high survival and prevalence rate. Both the tumor itself and systemic therapy can have an impact on patients’ nutrition. Conversely, preliminary data suggest that malnutrition has a negative impact on the development and prognosis of neuroendocrine tumors, as does obesity. The aim of this review is to condense the latest evidence on the role of the most widely used dietary patterns, the Mediterranean diet, the ketogenic diet and intermittent fasting, in the context of neuroendocrine tumors. Nutritional plans are an integral part of the multidisciplinary treatment team of patients with neuroendocrine tumors because they improve the patient’s quality of life. The nutritional approach must be tailored, based on nutritional needs and nutritionally manageable signs and/or symptoms related to drug treatment.

**Abstract:**

Neuroendocrine neoplasms are a heterogeneous group of neoplasms with increasing incidence, high prevalence, and survival worldwide. About 90% of cases are well differentiated forms, the so-called neuroendocrine tumors (NETs), with slow proliferation rates and prolonged survival but frequent development of liver metastases and endocrine syndromes. Both the tumor itself and systemic therapy may have an impact on patient nutrition. Malnutrition has a negative impact on outcome in patients with NETs, as well as obesity. In addition, obesity and metabolic syndrome have been shown to be risk factors for both the development and prognosis of NET. Therefore, dietary assessment based on body composition and lifestyle modifications should be an integral part of the treatment of NET patients. Nutrition plans, properly formulated by a dietician, are an integral part of the multidisciplinary treatment team for patients with NETs because they allow an improvement in quality of life, providing a tailored approach based on nutritional needs and nutritional manageable signs and/or symptoms related to pharmacological treatment. The aim of this review is to condense the latest evidence on the role of the most used dietary models, the Mediterranean diet, the ketogenic diet, and intermittent fasting, in the context of NETs, while considering the clinical and molecular mechanisms by which these dietary models act.

## 1. Introduction

Neuroendocrine neoplasms represent a group of tumors characterized by wide biological variability and clinical heterogeneity [1,2]. They arise from cells of the neuroendocrine system and can occur in all tissues and organs. However, the gastroenteropancreatic (GEP) and respiratory tracts are the most affected. Neuroendocrine tumors (NETs), which represent the most frequent type (about 90% of cases), are slowly growing and highly surviving tumors, but development of liver metastases, as well as endocrine syndromes is common [1,2]. In the last twenty years, a progressive increase in the incidence of neuroendocrine neoplasms has been reported, with it now being more than 6 cases per 100,000 per year [3,4]. This dramatic incidence increase can mostly be most explained by an improvement in the diagnostic ability, but specific risk factors cannot be excluded.

A few neuroendocrine neoplasms can create biogenic amines and peptide hormones, which may be related with clinical disorders, e.g., the carcinoid disorder (CS) caused by unmetabolized overproduced serotonin, hypoglycemic disorder in case of insulinoma, or Zollinger–Ellison disorder along with gastrinoma [1,2]. The treatment choices for neuroendocrine neoplasms incorporate surgery, somatostatin analogue therapy, radionuclide therapy, chemotherapy, molecular targeted therapies, alpha-interferon therapy, and inhibitors of serotonin production. In the case of hypersensitivity to biogenic amines, a diet that limits most the sources of amines ought to be utilized. The symptoms are ordinarily associated with histamine, tyramine, and putrescine [1,2]. As a result of the sickness, the utilization of a few nutrients can be diminished, driving to nutritional deficiencies, and resulting in malnutrition [5]. Furthermore, changes in digestion system may lead to cachexia in a few of these patients [6]. The crucial role of diet in the development, progression, and outcome of cancer is well known, and the consumption of healthy foods and nutrients is considered an effective strategy for primary cancer prevention, especially in gastrointestinal neoplasms, while tailoring the diet in the disease phase is the key to coping with the symptoms that cancer brings [7]. Based on this potential, several dietary patterns for patients with cancer are currently being studied.

Actually, the largest body of scientific work concerns the Mediterranean Diet (MD). MD’s beneficial effects on cancer are due to the high content of antioxidants and anti-inflammatory nutrients of Mediterranean food (wholegrains, vegetables, nuts, fresh fruit, fish, and extra virgin olive oil), which have a protective effect in counteracting cell degeneration and cancer cell proliferation [8,9,10]. In fact, there is evidence that low adherence to MD may influence the aggressiveness of various tumor types, such as prostate, bladder, and breast cancers [11,12], and, moreover, represent a dietary pattern suitable for both prevention and reduction in mortality from various cancers [10]. Following these observations, a cross-sectional case-control observational study investigated nutritional status, including adherence to MD, in 83 GEP-NET patients, compared to 83 healthy subjects, matched for age, gender, and body mass index (BMI) [13]. The results showed that, compared to healthy subjects, patients with GEP-NETs consumed Mediterranean food less frequently and Western foods (red/processed meats, butter, and sugar drinks) more frequently. Furthermore, NETs patients with more aggressive disease (G2 status) presented lower adherence to MD than patients with less aggressive disease (G1 status). These results highlighted in patients with NETs the association between nutritional status and tumor aggressiveness and the key role of an experienced dietitian in the multidisciplinary management of patients with NETs [13]. In fact, lower adherence to MD could also result from increased symptomatology that limits proper nutrition, representing an important aspect to take into account when applying dietary treatment.

Ketogenic diet (KD), a high-fat, low-carbohydrate diet with an adequate amount of protein, is now also considered a promising approach for the treatment of various cancers, particularly gynecological and neurological cancers [14]. It appears to sensitize most tumors to standard treatment by exploiting the reprogrammed metabolism of cancer cells, making it a promising candidate for adjuvant cancer therapy. Indeed, KD probably creates an unfavorable metabolic environment for cancer cells by inhibiting several signaling pathways involved in cancer cell survival [14]. Moreover, this dietary pattern has been shown to be safe and well tolerated [15,16] and to increase quality of life [17], but there is still a need for studies to confirm these results in NETs.

Finally, caloric restriction (CR) without malnutrition remains the most robust cancer prevention intervention to date in animal models [18,19], and in humans it promotes anticarcinogenic adaptations, such as reduced production of growth factors, inflammatory cytokines, and anabolic hormones, as well as reduced oxidative stress and free radical-induced DNA damage [18,19,20,21]. Despite a wealth of literature on the mechanisms and effects of CR, its clinical applicability remains limited because of challenges with long-term sustainability. Thus, newly designed dietary compositions aimed at inducing fasting-like effects that enable nutrition are beginning to emerge as potential therapies for delaying age-related diseases, such as cancer [22]. These fasting-like diets, such as intermittent fasting (IF), also promote the regeneration and rejuvenation of multiple systems by promoting stem cell self-renewal and white blood cell formation [22]. Therefore, novel, and periodic forms of fasting-like diets, as well as targeted reductions of specific macronutrients and a combination of both are likely to replace the original balanced and chronic restriction of all caloric sources.

Based on this background, we aimed to condense the latest evidence on the role of the most used dietary models (MD, KD, and IF) in the context of NETs while considering the clinical and molecular mechanisms by which these dietary models act.

## 2. Management of Tumors though Nutritional Approaches

### 2.1. Clinical Studies

In order to improve the quality life of cancer patient during his journey, several nutritional approaches have been evaluated before, during, and after treatments. The main goals of nutritional therapy for cancer patients are to prevent malnutrition and maintain a healthy weight, assist in symptom management, and reduce risk of treatment intolerance [23]. Indeed, international nutrition societies, such as the American Society of Clinical Oncology (ASCP), the American Society of Parenteral and Enteral Nutrition (ASPEN), the European Society for Clinical Nutrition and Metabolism (ESPEN), and others, have published clinical practice guidelines with recommendations concerning cancers’ patient nutrition care [24]. In general, these recommendations suggest dietary patterns adding value to fresh fruits, vegetables, proteins, and wholegrain cereals while proposing a restricted intake of fat, salt, sugar, and processed food [24]. Moreover, it has been demonstrated that dietary models based on a regular caloric intake of fruits, vegetables (especially cruciferous and garlic), and aliments rich in nutrients, such as vitamins (B-12 or D), folic acid, antioxidants, and selenium can play a protective role in cancer onset [25]. Whereas meat and animal products may increase the incidence of cancer [25]. In the past years, multiple dietary models have been developed following the above-mentioned international guidelines also considering the complex metabolism of cancer. It is estimated that changes in alimentary habits can contribute to avoid cancer onset of 30–50% [26,27]. Indeed, in a case-control study conducted in China on 337 case control pairs within the Shanghai Breast Cancer Study (SBCS), the intake of cruciferous vegetables measured by the urinary secretion of isothiocyanates was inversely related to the risk of breast cancer [28]. In addition, prostate cancer risk was found to be reduced by cruciferous vegetable consumption in a population-based case control study carried out in western Washington State [29]. Three or more servings per week, compared to less than one serving of cruciferous vegetables per week resulted in a statistically significant 41% decrease in prostate cancer risk [29].

Although the above-mentioned studies have reported promising results in cancer prevention, no dietary recommendations can be applied in active cancer patients or during cancer treatments. Indeed, in 2018 the third report of the World Cancer Research Fund/American Institute for Cancer Research (WCRF/AICR) concluded that the evidence of decreased cancer risk associated with greater adherence to dietary models was too scarce, except for cancers of the mouth, pharynx, and larynx [30]. The reason of this conclusion may be also addressed to the wild panel of dietary patterns involved in the studies without a homologous use of descriptive and analytics index [30].

Due to the high prevalence of preclinical evidence and to the strong rationale underlying the association between dietary patterns and cancer incidence, new clinical studies are necessary. Notably, clinical trials and epidemiological studies, together with randomized clinical trials need to be performed with more stringent and specific dietary patterns, in order to acquire consistent and reliable results. In this review, we will emphasize the use of the three major healthy diets models, such as MD, KD, and IF, to better understand their application in cancer prevention, including NET tumors.

#### 2.1.1. Mediterranean Diet

MD embraces a high consumption of fruits, vegetables, nuts, seeds, cereals, and legumes, associated with moderate intake of fish, olive oil, and alcohol beside a low supply of red meat and dairy products [31]. Actually, MD is recognized by United Nations Educational, Scientific and Cultural Organization (UNESCO) as intangible cultural heritage of humanity, and it was defined as a social traditional practice ranging from the landscape to the cuisine of seven Mediterranean countries [32]. The beneficial effects of MD are related to the high content of polyphenol, vitamins, flavonoids, and antioxidants contained in Mediterranean foods [31]. Indeed, their action results in anti-proliferations effects, inhibition of degeneration, DNA damages prevention and anti-inflammatory effects [33]. Due to the large panel of the Mediterranean foods linked to ancient traditions and lifestyles, MD is not a regimen of diet strictly imposed. This peculiarity turns to be an obstacle when applied in clinical research; therefore, in order to overcome discrepancies, different scoring systems and indexes have been developed [34]. In particular, a priori scoring system using the data of each subjects includes the Mediterranean Diet Score (MDS) and the Italian Mediterranean Index [35,36]. The MDS score has nine proper Mediterranean components [36], while the Italian Mediterranean Index includes eleven components: four non-Mediterranean foods, six typical Mediterranean foods, and alcohol [35]. On the other hand, a posteriori scoring is determined after a principal component analysis aimed at defining specific diet model, such as the modified Mediterranean diet (mMED) or alternate Mediterranean diet (aMED) [37,38]. A posteriori scoring system has been used in many observational and prospective studies directed to determine the incidence of each disease and the overall mortality rate [39]. Both, the mMED and aMED consist in a modified MDS, where the ratio of monounsaturated fatty acid (MUFA) + polyunsaturated fatty acid (PUFA) is used as lipid intake. In particular, mMED together with MUFA/PUFA ration measure the saturated fatty acid (SFA) component [37,38]. Notably, employing these methodologies, an inverse and significant correlation between MD and cardiovascular mortality rate was described [40].

Overall, these assumptions together with the well-known relationship between obesity and cancer risk [41,42,43,44], have paved the way to study the association of cancers incidence and mortality with adherence to MD. Although some studies showed that diets rich in fruits and vegetable were not efficient in the reduction in body weight, consistent proof of association between adherence to MD and patient’s body weight have been reported [42,43,44]. PREDIMED and PREDIMED-plus studies are two randomized clinical trials conducted in Spain on high-risk population that provided solid evidence of adherence to MD in weight control [43]. PREDIMED intervention involved subjects with an ad libitum (no dietary intervention) diet, whereas PREDIMED-plus study promoted both physical activity and energy-restricted MD [43]. Nowadays, the most important findings on the association between adherence to MD and cancer risk have been described, especially on breast cancer [45,46], colorectal cancer (CRC) [35,47,48], gastric cancer (GC) [49,50,51], and prostate cancer [52,53,54]. Considering both aMED (excluding alcohol) and MDS score, it emerged that the incidence risk of breast cancer significantly decreased (40%) for post-menopause women [46]. Moreover, it achieved 14–18% when medium or high adherence to MD, together with a moderate alcohol intake, was pursued [45], whereas CRC risk was reduced to about 30% and 45% in both men and women, respectively, with a high adherence to MD [47]. Notably, the predictive capacity of MD in the evaluation of the risk did not change, even when employing different MD scores, such as the Italian Mediterranean Index, mMDS, and center-specific mMDS (CSmMDS), or considering confounding variables [35,48]. Actually, when sex or age variables were considered, it was observing a reduction in mortality rate of 11% and 12%, respectively, for each 1-point increase in mMDS [48]. Concerning GC risk, two case-control studies were conducted in Italy between 1985 and 2007 [49,50]. These two studies were described by Praud and colleagues in an interestingly meta-analysis, in which they were able to assign a proper MDS score to each case control study food frequency questionnaire (FFQ) [51]. Therefore, comparing subjects in the lower category of adherence to MD (0–3) with those in the medium (4–5) and high [6] category, showed a 22% and 43% decreased risk of GC, respectively, hence demonstrating a significant and inverse trend between MD and GC risk [51]. Finally, in prostate cancer patients, it has been shown that a high adherence to MD was not only inversely associated with a low incidence of prostate cancer but also associated with decreased cancer malignancy (44%) and mortality in patients with metastasis (22%) [54]. Overall, a high MD score was associated with a low possibility of prostate cancer [52]. Therefore, prostate cancer risk decreased until 78% in subjects with the highest MD scores, registering in particular a decrease of 14% for each one-point increase in MD score [53].

Interestingly, a novel study of the University of Rochester Medical Center, New York started in 2020 (clinicaltrial.gov NCT04534738). This study will assess the feasibility of delivering an 8-week MD, as well as the efficacy of preliminary interventions on cancer-related fatigue among patients undergoing chemotherapy, compared to usual care (clinicaltrial.gov NCT04534738).

#### 2.1.2. Ketogenic Diets

KD is a high-fat, low-carbohydrate diet with adequate balance of protein and calories [25,55]. Initially, it was developed as therapeutic treatment for pharmacoresistant epilepsy; subsequently, due to its effect in lowering glucose plasma levels, it has been proposed to take advantage of altered metabolism in cancer [25]. Moreover, it has been reported that the low carbohydrate composition (<50 g) is associated with a decreased level of obesity [56,57,58], body weight, and inflammation [59]. These assumptions have grown the clinicians’ interest in the effect of KD application on cancer patients, and, nowadays, emerging results have been reported, as subsequently described. KD dietary pattern requires 65–90% of the daily energy as fat, thus the meals consist of foods enriched of butter, cream mayonnaise, or oils with eggs, meat, and fish—beside cheese—is highly recommended as a protein source [25]. In 1995, the beneficial effects of KD was shown in pediatric patients with advanced-stage malignant astrocytoma [60]. In this case report study, two female pediatric patients with advanced stage malignant astrocytoma tumors displayed an average decrease of 21.8% in glucose uptake after eight weeks of KD at the tumor side [60]. In addition, a clinical trial of 20 patients with recurrent glioblastoma showed a longer progression-free survival under the KD association with standard therapy [61]. Nowadays, clinical studies concerning safety, tolerability, and survival effects reported significant and promising results in several other cancers, including lung cancer [62], endometrial and ovarian cancer [63], breast cancer [16], pancreatic [64], and gastric cancers [65]. The feasibility study preformed on lung cancer demonstrated a high level of response and survival in 44 metastatic non-small cell lung cancer patients undergoing KD [62]. On the other hand, a randomized controlled trial on 73 endometrial and ovarian cancer patients showed a selective loss of fat mass together with a reduced serum insulin level and retention of lean body mass after 12 weeks of KD [63]. Notably, Cohen and colleagues, recording elevated b-hydroxybutyrate in patients’ serum, demonstrated anti-proliferation effects of KD [63]. Overall, these results suggest that KD may be safe tool for endometrial and ovarian cancer. On the other hand, concerning breast cancer, 60 patients treated with chemotherapy associated to KD showed safety, tolerability, and an improved survival, compared to the control group under a standard diet [16]. Whereas in a retrospective observational study with 25 metastatic pancreatic ductal carcinoma patients included, KD compared to the general diet displayed an improved meal compliance, energy intake rate, and meal satisfaction [64]. Finally, survival outcomes were improved in 24 patients with stage III-IV advanced or metastatic gastric adenocarcinoma [65]. Patients received KD combined with chemotherapy, local hyperthermia, and hyperbaric oxygen therapy [65]. These findings suggest that KD is able to promote an improper metabolic environment for tumor cells, assuming a possible role as promising adjuvant for patient-specific multifactorial therapy.

#### 2.1.3. Intermittent Fasting

IF represents one of the strategies to CR and is becoming a popular alternative to daily CR. Indeed, according to the International Food Information Council survey, IF is the most frequently cited diet pattern in 2020 among Americans aged 18 to 80, according to the International Food Information Council (IFIC) [66]. The IF dietary model is characterized by episodic periods of little to no calorie consumption with variation that can range from every-other-day complete 24 h fasting to fasting on 1 or 2 non-consecutive days per week (referred as the 6:1 and 5:2, respectively) [67]. Many fasting programs recommend no or small caloric intake during the fasting period together with an unlimited number of calorie-free beverages, such as water, coffee (without sugar or milk), bone broth, and diet soft drinks [68,69]. The most popular approach is the alternate-day fasting (ADF) in which meals can be either high or low fat if they are less than 500 calories; another variation of IF is named time restricted feeding (TFR) and requires a limited consumption of calories, typically between 4 and 12 h daily [70]. Although initial concern about the fasting exacerbation in cancer patients leading to malnutrition, no significant results were reported in clinical studies [20,67]. The promising effect of both ADF or TRF on glucose, oxidative stress, leptin, and adiponectin implicated in cancer pathogenesis has been reported largely in the literature [71,72,73,74,75]. However, studies of IF adherence in cancer population are limited and are mostly still on trial (recruiting or active phase). Notably, a small nonrandomized study of 23 women at high risk of breast cancer showed that IF adherence of 1 month (2 days per week with 65% of energy restriction) led to a 4.8% reduction in body weight, an 8% reduction in body fat, and an improvement in the homeostatic model assessment (HOMA) for insulin resistance index on both fasting and non-fasting days [76]. Due to this initial result of IF adherence on breast gene expression and breast cancer risk factors, different studies has started analyzing the association between IF, tumor risk factors, and tumor progression. The on-going clinical trials include colon cancer patients (clinicaltrial.gov NCT04345978), gliomas (clinicaltrail.gov NCT04461938), chronic or small lymphocytic (clinicaltrial.gov NCT04626843), childhood cancer (clinicaltrial.gov NCT04626843), and prostate cancer patients (clinicaltrial.gov NCT04288336). Notably, the role of IF in tumor progression is currently studied in patients treated for metastatic breast and prostate cancer with neuroendocrine therapy (clinicaltrail.gov NCT04708860; clinicaltrial.gov NCT02710721).

### 2.2. Molecular Mechanisms

#### 2.2.1. Mediterranean Diet

MD can favorably contribute to anticancer treatment by several mechanisms [77]. It reduces tumor cell growth by acting on hormones and extra and intracellular pathways, has anti-inflammatory, anti-oxidative and anti-aggregating effects, modulates hormones and growth factors involved in cancer, reduces lipids, and positively affects the production of microbiota bacterial metabolites [77]. The anti-tumoral effects of MD can be played by many food compounds, including fish, fruit, vegetables, cereals, olive oil, and legumes [78]. These elements are full of antioxidants, such as vitamins A, C and E, carotenoids, lycopene, of flavonoids, polyunsaturated fatty acids (PUFAs) and fibers, which are involved in many cellular signaling pathways [78]. Antioxidants scavenge reactive oxygen species (ROS) and reduce their production. ROS are involved in the stimulation of phosphoinositol-3-kinase (PI3K) and mitogen-activated protein kinases (MAPK), which in turn activate the nuclear factor kappa beta (NF-kβ) favoring cancer development [79,80,81,82]. Flavonoids directly reduce the activation of PI3K and MAPK pathways; inhibit the conversion of pro-carcinogens in active carcinogens; increase the expression of phase 2 enzymes, involved in the catabolism of carcinogens; stimulate the AMP-activated protein kinase, which, in turn, is involved in the modulation of NF-kβ activity; moderation of cell cycle; and favor DNA-repairing molecules [83]. Fibers act on gut microbiota, releasing short-chain fatty acids (SCFAs), which can reduce the expression of NF-kβ in enterocytes, reduce the proliferation of cancerous colonocytes and reduce the expression of local neutrophils, lymphocytes, and dendritic cells [84,85]. Further, they reduce the insulin and IGF-1 secretions, which are involved in tumor growth [86,87]. PUFAs contributes to reduce the chronic low-grade inflammation, by down-regulation of NF-kβ and decreased expression of inflammatory cytokines, including IL-6, IL-1β, and TNF-α, and anti-angiogenetic, such as VEGF [88]. Red wine is a source of resveratrol and quercetin able to modulate cell cycle, induce apoptosis in cancer cells, and act as an anti-inflammatory [89]. However, the beneficial effects of red wine are controversial [89].

#### 2.2.2. Ketogenic Diet

KD has been reported as an effective anticancer therapy, enhancing chemotherapy effect, reducing inflammation, and regulating the expression of many factors and proteins, including matrix metalloproteinases, histone deacetylases, AMP-activated protein kinase, pyruvate kinase (AMPK), and p53 [14,90]. The anticancer effects are played both by a decrease in glucose levels and by the production of ketone bodies. KD reduces glucose uptake, essential for tumor cells, which are glucose dependent (Warburg effect) and increases ketone bodies, which cannot be used by cancer cells [91,92]. Interestingly, KD reducing carbohydrate assumption reduces insulin and IGF-1 levels and increases insulin sensitivity. Low IGF-1 levels result in the inhibition of PI3K/Akt/mTOR system and of the synthesis of vascular endothelial growth factor (VEGF), which are involved in angiogenesis, cancer cell proliferation, and survival [93,94,95]. KD reduces the production of pro-inflammatory cytokines, including tumor necrosis factor-α (TNF-α), interleukin-1β (IL-1β), interferon-γ (IFN-γ), and cyclo-oxygenase 2 (COX-2), which are involved in tumor growth, proliferation, angiogenesis, invasion, metastasis, and DNA damage [96,97]. In addition, KD decreases the expression of matrix metallo-proteinases-9 and inhibits the expression of histone deacetylases, resulting in a modulation of the methylation status of oncogenes and, consequently, in a tumor growth slowdown and induction of apoptosis [98]. Further, KD inhibits the expression of pyruvate kinases M2 (PKM2) [87]. PKM2 is involved in the process of glycolysis, catalyzing the conversion of phosphoenolpyruvate to pyruvate with the production of ATP, and it is strongly expressed in cancer cells, mediating their metabolism [99]. Another interesting effect of KD is the silencing of p53 mutant factor. In cancer cells, there are mutations in p53 factor favoring the cancer cell proliferation [88]. KD, reducing glucose levels, results in increased deacetylation and degradation of p53 mutant, reducing its expression in cancer cells [100]. In the end, KD reducing glucose uptake increases the expression of AMPK, resulting in the activation of p53 factor and further inhibition of mTOR expression resulting in the inhibition of tumor growth and proliferation and stimulation of cancer cells apoptosis [101,102] (Figure 1).

Ketogenic diet could be an effective adjuvant anticancer therapy, reducing glucose uptake (Warburg effect) and production of pro-inflammatory cytokines (TNFα, IL-1β, IFN-γ, COX-2) and regulating the expression of many factors and proteins, including matrix metalloproteinases, histone deacetylases, pyruvate kinase, p53, and AMP-activated protein kinase, involved in tumor growth and progression. Abbreviations: MMP9, matrix metallo-proteinases-9; HDAC, histone deacetylases; PKM9, pyruvate kinases M2; AMP-K, pyruvate kinase; TNFα, tumor necrosis factor-α; IL-1β, interleukin-1β; IFN-γ, interferon-γ (IFN-γ); COX-2, cyclo-oxygenase 2; (→), activation; (┬), inhibition.

#### 2.2.3. Intermittent Fasting

The effects of IF in cancer have been demonstrated in preclinical rodent studies [103]. IF was shown to reduce blood glucose, leptin, insulin, and IGF-1 levels, leading to the modulation of the PI3K/AKT/mTOR pathways [104,105]. Notably, IF is associated with an increase in insulin-like growth factor-binding protein (IGFBP-1), reducing IGF-1 bioavailability, and thus reducing cell motility and metastasis [106,107]. In addition, IF was associated with ketone body formation that blocks histone deacetylases and, consequently, tumor growth, and, at the same time, plays an antioxidant function [108,109]. Further, IF could stimulate cellular autophagy, resulting in the elimination of damaged molecules and proteins [110]. Autophagy can act as an anticarcinogenic mechanism against tumor transformation. In addition, autophagy may act to increase the sensitivity to chemotherapy and reduce the toxicity [20]. Finally, IF reduces inflammation by decrease in C-reactive protein, IL-6 and TNF-α [111]. However, all the above-mentioned data are available from animal studies and need to be confirmed in humans.

## 3. Nutrition and Symptoms Related to Neuroendocrine Neoplasms

Unhealthy diets, along with visceral fat accumulation, could cause cancers, including NETs [112]. In this regard, Barrea et al. found an association between high visceral adiposity index (VAI), which is a gender-specific indicator of adipose dysfunction, high fatty liver index (FLI), a tool for identifying individuals with non-alcoholic fatty liver disease, and metabolic syndrome, with worse clinicopathological characteristics in NET patients [113]. On the other hand, several studies underlined the risk of malnutrition in patients with NET [5,6,114,115]. Specifically, Maasberg et al. highlighted that about 20–25% of patients with NET were at risk of manifest malnutrition, determined by biochemical analysis, anthropometric measurements, and bioelectrical impedance analysis (BIA) [5]; similar results were described by Laing et al. [114] and Borre et al. [6] in their cross-sectional studies, who found 29% and 25% of patients with NET were malnourished, respectively. A lower prevalence of malnutrition was assessed by Qureshi et al., who found this condition in 14% of outpatients with GEP-NET [115]. However, the authors recommended a routine use of malnutrition screening in all patients with GEP-NET.

Furthermore, a worse metabolic profile in NET patients is also associated with higher ki67% index and larger tumor size [116]. For these reasons, nutrition assessment and management should be an integral part of the management of patients with NETs and should be performed by an accredited dietitian [116,117]. Interestingly, Leoncini et al. conducted a meta-analysis and highlighted that body mass index (BMI) was the most relevant risk factor for NETs development, only after family history of cancer [118]. On the other hand, malnutrition could negatively influence the quality of life and survival [5]. Nutritional status in NET patients, mainly gastrointestinal NET, is also affected by the increased production of gastrointestinal hormones, peptides, and amines [116]. In this regard, the clinical manifestations of NETs could be characterized by various hormone hypersecretion syndromes, such as the CS or the hyperinsulinemic hypoglycemia syndrome, in case of insulinomas [116].

Diet should be formulated and individualized by an accredited dietitian, according to the patients’ characteristics, considering age, stage of the disease, pharmacological and/or surgical treatment, and symptoms (i.e., diarrhea or constipation, hypoglycemia, or hyperglycemia) [116,119].

### 3.1. Diet Protocol and Neuroendocrine Tumors

#### 3.1.1. Mediterranean Diet

MD is considered one of the healthiest dietary patterns, and there is a correlation between low adherence to MD and cancer, and adherence to MD could influence the aggressiveness of different tumor types (i.e., prostate, breast cancer, and melanoma) [12,120]. For all these reasons, MD could be considered a useful model to adopt in NET patients.

Barrea et al. conducted a case-control, cross-sectional study to investigate the nutritional status in a selected group of GEP-NET patients and the adherence to MD, using a validated 14-item questionnaire for the assessment of PREvención con DIeta MEDiterránea (PREDIMED) [13]. The authors found that patients with more aggressive disease showed a lower adherence to MD, compared to G1, localized and free/stable disease status; furthermore, the presence of metastases and progressive disease were higher in patients with scarce adherence to MD. This study highlighted the relationship between nutritional status and tumor aggressiveness, showing how MD could represent an important tool for the prevention of cancer aggressiveness in patients with NET [13]. Obviously, the strength of evidence is related only to one study and further studies are needed to confirm this finding.

Due to the importance of adherence to MD, which appear to be a dietary pattern suitable for the prevention of the disease, some recommendations have been proposed in patients without current disease [121]: (a) to perform a nutritional dietary planning in all NET patient; (b) to limit consumption of red and processed meat, sugar, and sweet beverages; (c) to avoid alcohol, and d) to prefer fruit and vegetables, legumes, and grains.

In patients with CS, small and frequent meals should be suggested [121].

#### 3.1.2. Ketogenic Diet

KD regimen showed promising results as an adjuvant cancer therapy, mainly targeting tumor metabolism, through the modulation of gene expression, as well as tumor microenvironment. To date, there is a lack of randomized controlled trials on the effect of KD on cancer in general [122,123] and, specifically, in NET patients. Moreover, KD, inducing the body to get energy from fat instead of glucose, could antagonize the “Warburg effect”, which refers to the aerobic glycolysis that occurs in cancer cells [124]. This hypothesis leads us to speculate that KD can improve the condition of insulin resistance, even before weight loss. In fact, KD increased the insulin sensitivity in patients suffering with type 2 diabetes mellitus by about 75% [125]. This effect may be particularly useful in some forms of secreting NETs associated with hyperinsulinemia [126]. Interestingly, Fine et al. conducted a pilot study with end-stage cancer and observed that 28 days of KD were related to a significant reduction in insulin levels and a better therapeutic response [127].

On the other hand, it should also be considered that a high-fat diet can worsen or induce symptoms related to fat malabsorption; therefore, caution is needed in KD use.

#### 3.1.3. Intermittent Fasting Diet

In the last decades, a growing interest has been addressed to IF, including different strategies (mainly one or two non-consecutive days per week of 24 h fasting) [67]. Even if preclinical studies showed the promising effects of IF on cancer [128,129], to date, no studies conducted in NET patients are available. However, based on studies in non-cancer patients, we can only speculate that the anti-inflammatory effects and overweight and obesity reduction related to IF could also improve the tumor responses to anticancer treatment in NET patients [130]. Furthermore, it has been supposed that IF could induce an anti-Warburg effect, similar to KD [128]. Obviously, this is an assumption without enough evidence to make recommendations, and further studies are needed.

### 3.2. Management of Symptoms through Nutritional Approaches

#### 3.2.1. Diarrhea

Diarrhea is a frequent symptom in patients with NET. The main causes are tumor growth and mass effects, as well as over secretion of peptides and hormones by functioning tumors [131]. Furthermore, diarrhea and steatorrhea are common adverse effects of SSAs, due to treatment-induced pancreatic insufficiency [131]. It would be advisable, extending the evidence deriving from non-NET patients, to avoid sources of histamine, such as baker’s yeast, wine, beer, and cheese, as well as highly processed cold meat [132]. In patients with CS, diarrhea, mainly secretory, occurs in 80% of cases. In these cases, frequent small meals and avoiding aged cheeses and fermented foods with high amine content could help to reduce symptoms; furthermore, these indications still require strong scientific evidence. Moreover, replenishing liquids is essential [133].

MD, high in fruit, vegetables, and grains, represents a good dietary pattern; however, patients with symptomatic NETs should reduce insoluble fiber and increase soluble fiber consumption [116]. Moreover, studies conducted on healthy subjects found that adherence to MD dietary pattern could improve fecal microbiota composition [134]; this concept could also be extended to NETs patients. Regarding the effects of KD on diarrhea, a cross-sectional study conducted in 17 patients with diarrhea-predominant irritable bowel syndrome showed an improvement in abdominal pain, stool habits, and quality of life after four weeks of this diet regimen [135]; these results could open a new scenario in the management of patients with NET. Lastly, IF seems to reduce gastrointestinal discomfort related to diarrhea in cancer patients [136], and this could also be helpful in NET patients; however, limited numbers of human studies are available.

#### 3.2.2. Constipation

Chronic constipation can occur in patients with NET less frequently than diarrhea, mainly due to side effects of treatment, ileus, or catecholamines excess; flatulence, pain, and abdominal distension are often related symptoms [137]. In this contest, along with physical exercise, diets should prevent constipation through liquids consumption (i.e., mineral water, juice) and following a high-fiber diet (whole grains, vegetables, and fruit with high fiber content) [116]; for these reasons, MD could represent a valid dietary pattern. KD should be used with caution, based on studies that report an increased risk of constipation in patients using this nutritional pattern, especially those over 70 years of age [138]. Finally, general indications suggested not to skip meals [139]; in this context, IF regimen should be avoided.

#### 3.2.3. Malnutrition and/or Cachexia

Malnutrition, which is a deficiency or imbalances in a person’s intake of energy and/or nutrients, and cachexia, which is characterized by a loss of fat and muscle mass, frequently occur in cancer patients [140]. The main causes are: (1) anorectic effects of the tumor, and (2) a limited food consumption, which is frequently caused by nausea, vomiting, pain, and inflammation of the mucous membrane related to anticancer treatment [140]. The consequences are body mass loss and tiredness [141]. In this contest, a well-balanced diet, i.e., MD, should prevent body mass loss, including an appropriate amount of all nutrients (proteins, fat, carbohydrates, vitamins, and minerals) and consuming meals more frequently. Supplementation with omega-3 fatty acids, as well as arginine may support the nutritional profile. In patients with CS, niacin (vitamin B3) deficiency can occur due to the increased metabolism of tryptophan (which is the precursor of niacin) into serotonin; vitamin B3 supplementation could be helpful [142].

KD used in patients with cancer usually obtain weight loss [90]; however, it has been demonstrated that in cachectic patients, KD induced weight gain [143]. These issues require further studies in order to find effective clinical indications.

#### 3.2.4. Glycemic Imbalance: Hypoglycemia and Hyperglycemia

Hormones hypersecretion could be a cause of imbalanced nutrition and metabolic changes in patients [144]. Patients with insulinoma showed the low level of glucose and the role of diet is to prevent long fasting between meals, as well as the supply of high glycemic index carbohydrates in the case of hypoglycemia. For these reasons, IF diet should be avoided. Glucose metabolism can also be slowed down by a high-protein diet [144]; the high protein content of KD may be supportive, but the absence of carbohydrates should be avoided too. Low glycemic index and complex carbohydrates should be preferred, while rapidly absorbable carbohydrates with a high glycemic index are suited during hypoglycemic episodes [116].

On the other hand, hyperglycemia can occur as an effect of some pharmacological treatment (SSAs), surgery (pancreatectomy with subsequent iatrogenic diabetes), or tumor mass, causing a reduction in insulin secretion [1]. A healthy balanced diet, with reduced carbohydrate intake and high consumption of fiber could be helpful to prevent or delay the onset of diabetes, as well as to improve glycemic control [145]. In this regard, MD represents a healthful eating pattern, which can improve glycemic profile, and delay the need for antidiabetic drug therapy, as also recommended by the American Diabetes Association (ADA) guidelines [146]. On the other hand, KD may represent an alternative, due to insulin-sensitizing effects, reduction in blood glucose, and insulin-like growth factor, which are important drivers of cancer cell proliferation; similarly, IF protocol could ameliorate glucose metabolism, maintain the BMI, and reduce inflammation [147].

#### 3.2.5. NET Treatment-Related Symptoms

NET pharmacological treatment mainly includes SSAs and, in case of unresectable or metastatic NETs or neuroendocrine carcinoma (NEC), targeted therapies (*everolimus and sunitinib*) and chemotherapy [4]. SSAs, which are the most common pharmacological treatment for advanced well-differentiated NETs, can induce exocrine pancreatic insufficiency, resulting in diarrhea and steatorrhea [148]. Considering target therapy, some foods (i.e., grapefruit, chamomile, garlic, and ginseng) could inhibit cytochrome P450 3A pathway, with a subsequent increasing toxicity of this treatment [112]. With regard to temozolomide, patients should be instructed not to take it with fatty foods, as these can modify the stomach pH, inhibiting cytochrome P450; high-fat meals are also inhibitors of tyrosine kinase [148]. Chemotherapy, including fluoropyrimidine, oxaliplatin, and irinotecan, could induce diarrhea.

MD, which represents a healthy eating model, can be useful in the management of gastrointestinal adverse effects related to NET treatments. Furthermore, it could be speculated that KD can increase the effect of pharmacological treatment, acting on IL-6, VEGF, and PI3K/AKT/mTOR pathways [100]; this strengthens the idea that KD could be considered an “adjuvant cancer therapy”, translatable in the management of NETs. Finally, it could be supposed that the IF diet is capable of decreasing tumor growth, as well as chemotherapy-related toxicity, mainly diarrhea [149]; however, further studies on human are needed to investigate the safety and feasibility of fasting.

## 4. Conclusions

Neuroendocrine neoplasms and their main subtype NETs have an increasing incidence worldwide, associated with a high survival and prevalence rate. Surgery remains the only curative treatment for early stage disease, while SSAs are the treatment of choice for unresectable/advanced disease, followed by peptide receptor-targeted radiotherapy and various drugs, such as targeted therapy and chemotherapy. Both the tumor itself and systemic therapy may have an impact on patients’ nutrition. Malnutrition has a negative impact on the development and prognosis of NETs, as well as obesity. However, data on nutrition in NETs are scattered and large epidemiological studies, as well as randomized clinical trials are lacking. In addition to the well-known MD, new nutritional approaches seem to have important benefits for patients with NETs, such as KD and IF, and could be used appropriately according to patient characteristics. Therefore, dietary assessment based on body composition and lifestyle modifications should be an integral part of the treatment of NET patients. Nutrition plans are an integral part of the multidisciplinary treatment team for patients with NETs because they allow an improvement in the quality of life, providing a tailored approach based on nutritional needs and nutritional manageable signs and/or symptoms related to pharmacological treatment.

## Figures and Tables

**Figure 1 cancers-14-04402-f001:**
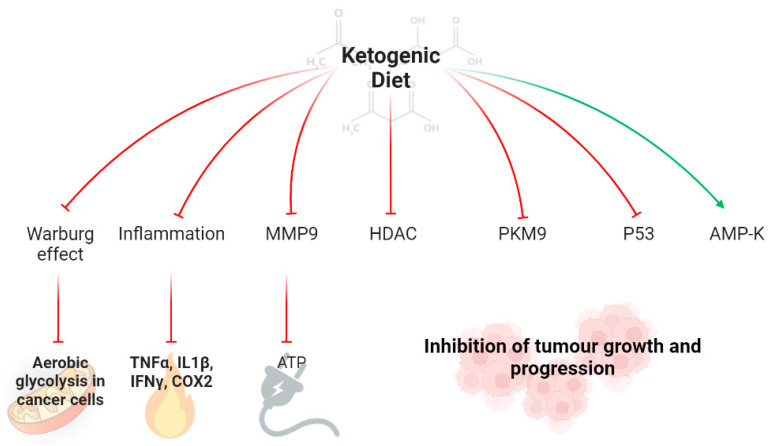
Anticancer effects of ketogenic diets.

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
