# Peer review of "Neuroendocrine Tumors: A Comprehensive Review on Nutritional Approaches"

_cancers, 2022, doi:10.3390/cancers14184402_

Round 1

Reviewer 1 Report

Summary:

Thank you for the invitation to review this paper, titled 'Neuroendocrine tumors: a comprehensive review on nutritional approaches'. This is a review paper summarising evidence for alternative diets (mediterranean diet, intermittent fasting, and ketogenic diet) and their applicability to neuroendocrine tumors. The authors have done well to describe the current evidence available for these diets in other cancer and non-cancer populations, including detailed information on the metabolic mechanisms of these diets on cells. I do however, have concerns about the recommendations in the article for use of these diets in NET patients and cancer patients in general. The majority of the studies examining ketogenic and fasting diets are preliminary or in pre-clinical models, and yet to be translated to the clinical setting or deemed safe and effective in a broader cancer population. Also most of the study interventions are applied to a cancer prevention setting and should not be translated to patients with active cancer/undergoing treatment unless the evidence indicates so. In clinical practice, there are no formal guidelines that recommend ketogenic diets or intermittent fasting for any cancer patient, due to lack of sufficient evidence, so the authors should make sure this is clear to the reader throughout the article, so that results are not misinterpreted. There also needs to be further context regarding the nutritional issues faced by patients with a NET (particularly at the start of the article), and the cause of these, so that there is better context when deciding if the proposed diets are appropriate in a NET population.

Some further specific comments:

  • The paper is quite long, I would in general reduce the word count if possible, especially given evidence for most aspects is not strong
  • Page 1, line 24: there are only a few studies that have looked at malnutrition and prognosis of NETs, so this needs to be re-worded to reflect this (it is not confirmed in research yet)
  • Page 1, line 27: 'weight excess' do you mean 'obesity'?
  • Page 1, line 40: please specifically mention the role of an accredited Dietitian in assessing diet and providing individualised nutrition plans for patients
  • Introduction (page 2): More emphasis should be put on NETs and their common symptoms/treatments, and nutritional concerns before discussing the different diets for evaluation in this review
  • Page 2, line 59-62: Need to tease out the differences in the role of diet in the prevention phase (pre-cancer) and at diagnosis and during treatment phases, because the role of diet and nutritional requirements vary widely accross these stages. The evidence and recommendations for the general population to prevent cancer are extremely different to those for patients with current disease and undergoing cancer treatment
  • Page 2, line 63: please state/summarise in which populations this research was undertaken
  • Page 2, line 71 onwards: When discussing results of this study looking at MD diet adherance in NETs, the authors should consider that lower MD adherance is likely due to increased symptoms and unwell status of patients who are therefore limited in ability to eat. This needs to be discussed as it has important clinical implications for applicability of this diet.
  • Page 2, line 78: I don't know if you can state that results of this study indicate that a MD diet can prevent tumour agressiveness, the evidence needs to be stronger and from multiple studies to make that association - please remove.
  • Page 2, line 83 onwards: I would advise being careful about interpreting results from studies done with obese (non-cancer) populations and those in breast cancer populations, which have cancer and symptom profiles very different to NETs. This either needs to be changed or additional context provided to compare to NETs.
  • Page 3, line 111: Please change to - "The main goals of nutritional therapy for cancer patients are to prevent malnutrition and maintain a healthy weight, assist in symptom management, and reduce risk of treatment intolerance"
  • Page 2, section 2.1: This section needs more emphasis on the context of the research/studies in question and that most of them discuss nutritional interventions in cancer prevention, which is a very different setting to during cancer treatment/active cancer. The recommendations for diet for cancer prevention essentially can not be applied to patients with active cancer.
  • Page 4: please summarise more, reduce words
  • Page 5, section 2.1.2: Caution needs to be taken when revising this section, as it currently reads that a ketogenic diet is effective and recommended in cancer populations, which in practice is not true. The evidence/research is still emerging and is limited to specific populations (not in any NET populations) so it would be important for the authors to make this more clear to the reader
  • Page 5, line 259: 'Caloric restriction is a well known diet intervention against cancer' - I would disagree with this and suggest that the evidence is not strong enough to make this recommendation. So please remove or re-word. Also please note that on page 9, line 2.2.3 the authors write 'data (on intermittent fasting) are available from animal studies and need to be confirmed in humans' this is great to include and important context. So worth ensuring this message is continued throughout the manuscript.
  • Page 6, line 274 and line 277: Both mention lack of evidence in clinical studies and that few studies have been done in a cancer population. If this is the case then why are the authors discussing these studies in great detail. I feel more emphasis needs to be made for the state of evidence and recommendations for where gaps are, and what we are unable to apply to cancer or NETs until further research is done.
  • Page 6 + 7, sections 2.2.1 and 2.2.2: please simply as able, the level of detail provided is good but probably not required in context of the article.
  • Page 9, section 3: There are additional studies looking at malnutrition prevalence in NETs, so please include: Laing et al. 2022, Borre et al. 2018, Qureshi et al. 2016.
  • Page 9, line 386-7 and line 393-395: Please add that the role of an accredited Dietitian is important to provide nutrition assessment and management.
  • Page 9, line 390-392: Please expand on how hormone production affects nutritional status of NETs, it is due to symptoms arising from hormonal syndromes.
  • Page 9, section 3.1.1 and 3.1.2: the introductory information included here is repetitive with previous sections so can reduce
  • Page 10, line 412: More emphasis needs to be placed on the strength of evidence here. I'm not sure you can say that a MD may prevent cancer agressivness after results of one study?
  • Page 10, line 414-418: Please adjust wording so it is clear these recommendations are for cancer prevention and not for patients with current disease
  • Page 10, line 434: Reference 126 cited with "this effect may be particularly useful in some form of secreting NETs...." is not relevant to glucagonomas (the article is regarding glucagon in general) and should be taken out along with the text.
  • Page 10, section 3.1.2: The authors should note that a high fat diet can worsen or induce symptoms in NET patients, as often they are suffering from fat malabsorption, so please re-word this section with this consideration.
  • Page 10, section 3.1.3: Please re-word any statements that indicate that an intermittent fasting diet may be beneficial or recommended for NET patients, as there is not enough evidence to make this recommendation.
  • Page 10, line 453: Please change to "due to treatment-induced pancreatic insufficiency"
  • Page 10, line 453: "It would be advisable to avoid..." this statement refers to a study that was not done in NET patients, so please ensure this is clear.
  • Page 10, line 455 onwards: Much of the evidence for avoiding amine foods is anecdotal, please ensure this is stated
  • Page 10, line 461: "assumption" change to "consumption"
  • Page 11, line 482: malnutrition and cachexia are different with different definitions, please revise so it's clear
  • Page 11, section 3.2.5: Should this read 'NET treatment-related symptoms'?
  • Conclusion, line 547-549: "new nutritional approaches seem to have important benefits for patients with NET, such as KD and IF, and could be used appropriately according to patient characteristics" - as recommended to authors previously, I am concerned about recommending these restrictive diets with such certainly when the evidence available is not in NET-populations and is limited. So please remove from the conclusion or re-word to ensure the reader knows the limitations to the evidence.

Author Response

Rebuttal letter

On behalf of all the authors, I would like to kindly acknowledge the Reviewer for the constructive comments on the manuscript. The modified text has been marked in red in the revised version of the manuscript. I respectfully trust that the changes we have made to the text could have contributed to ensure clarity to our findings and to have answered the pertinent and appropriate comments, as kindly requested. Thank you very much and our best regards.

Sincerely yours,

Dr Giovanna Muscogiuri

Page 1, line 24: there are only a few studies that have looked at malnutrition and prognosis of NETs, so this needs to be re-worded to reflect this (it is not confirmed in research yet)

  1. We agree with the reviewer and we have reworded the text to highlight this aspect.

Page 1, line 27: 'weight excess' do you mean 'obesity'?

  1. We thank the reviewer and have modified the text.

Page 1, line 40: please specifically mention the role of an accredited Dietitian in assessing diet and providing individualised nutrition plans for patients

  1. We agree with the reviewer and mentioned the role of the dietician in the multidisciplinary team.

Introduction (page 2): More emphasis should be put on NETs and their common symptoms/treatments, and nutritional concerns before discussing the different diets for evaluation in this review

  1. We agree with the reviewer and emphasised the suggested topics

Page 2, line 59-62: Need to tease out the differences in the role of diet in the prevention phase (pre-cancer) and at diagnosis and during treatment phases, because the role of diet and nutritional requirements vary widely across these stages. The evidence and recommendations for the general population to prevent cancer are extremely different to those for patients with current disease and undergoing cancer treatment

R: We agree with the reviewer and have better specified the role of nutrition in the various stages of the disease

Page 2, line 63: please state/summarise in which populations this research was undertaken

  1. Done

Page 2, line 71 onwards: When discussing results of this study looking at MD diet adherence in NETs, the authors should consider that lower MD adherence is likely due to increased symptoms and unwell status of patients who are therefore limited in ability to eat. This needs to be discussed as it has important clinical implications for applicability of this diet.

  1. We thank the reviewer for his insightful observation and have modified the text accordingly.

Page 2, line 78: I don't know if you can state that results of this study indicate that a MD diet can prevent tumour agressiveness, the evidence needs to be stronger and from multiple studies to make that association - please remove.

  1. We thank the reviewer and we removed that part.

Page 2, line 83 onwards: I would advise being careful about interpreting results from studies done with obese (non-cancer) populations and those in breast cancer populations, which have cancer and symptom profiles very different to NETs. This either needs to be changed or additional context provided to compare to NETs.

  1. We thank the reviewer for the remark and specified the lack of studies with similar results in NETs

Page 3, line 111: Please change to - "The main goals of nutritional therapy for cancer patients are to prevent malnutrition and maintain a healthy weight, assist in symptom management, and reduce risk of treatment intolerance"

  1. Thank you for your suggestion. We changed the text as indicated.

Page 2, section 2.1: This section needs more emphasis on the context of the research/studies in question and that most of them discuss nutritional interventions in cancer prevention, which is a very different setting to during cancer treatment/active cancer. The recommendations for diet for cancer prevention essentially can not be applied to patients with active cancer.

  1. We thank the reviewer for this observation. We added the following sentence “Although the above-mentioned studies have reported promising results in cancer prevention, no dietary recommendations can been applied in active cancer patients or during cancer treatments”. Moreover, at the end of the 2.1 section we changed “cancer treatment” with “cancer prevention”

Page 4: please summarise more, reduce word.

  1. We did summarize the indicated part as kindly suggested.

Page 5, section 2.1.2: Caution needs to be taken when revising this section, as it currently reads that a ketogenic diet is effective and recommended in cancer populations, which in practice is not true. The evidence/research is still emerging and is limited to specific populations (not in any NET populations) so it would be important for the authors to make this more clear to the reader.

  1. Thank you for this suggestion. We added the sentence “These assumptions have grown the clinicians’ interest in the effect of KD application on cancer patients and nowadays emerging results have been reported, as subsequently described.”

Page 5, line 259: 'Caloric restriction is a well known diet intervention against cancer' - I would disagree with this and suggest that the evidence is not strong enough to make this recommendation. So please remove or re-word. Also please note that on page 9, line 2.2.3 the authors write 'data (on intermittent fasting) are available from animal studies and need to be confirmed in humans' this is great to include and important context. So worth ensuring this message is continued throughout the manuscript.

  1. We thank the reviewer for this suggestion. We decided to remove the evidence.

Page 6, line 274 and line 277: Both mention lack of evidence in clinical studies and that few studies have been done in a cancer population. If this is the case then why are the authors discussing these studies in great detail. I feel more emphasis needs to be made for the state of evidence and recommendations for where gaps are, and what we are unable to apply to cancer or NETs until further research is done.

  1. Thank you for this kind suggestion. In this part, the intention of the authors is to present the two different approaches of IF (ADF and TRF) reporting evidences of association between IF and different key factors also involved carcinogenesis. Subsequently, authors presented in details the only clinical study reported on cancer patient population and finally listed the studies on active or recruiting phase. Due to the lack of results of these novel studies under examination, is quite difficult to drawn a line of suggestion. Moreover, the NET part is discussed in detail below in the manuscript. However, to make this part more clear, authors added the sentence “However, studies of IF adherence in cancer population are limited and mostly still on trial (recruiting or active phase)” and some words such as “risk factors including” were removed. Moreover, we rephrased the last part of the 2.1.3 section.

Page 6+7, sections 2.2.1 and 2.2.2: please simply as able, the level of detail provided is good but probably not required in context of the article.

  1. Thanks for your comment. We tried to shorten these two sections as you kindly suggested.

Page 9, section 3: There are additional studies looking at malnutrition prevalence in NETs, so please include: Laing et al. 2022, Borre et al. 2018, Qureshi et al. 2016.

  1. Thank you for your suggestion. We now included also these studies changing the paragraph as follows: “On the other hand, several studies underlined the risk of malnutrition in patients with NET (115 – 35415851 - 29373941 - 27147385). Specifically, Maasberg et al. highlighted that about 20-25% of patients with NET were at risk of manifest malnutrition, determined by biochemical analysis, anthropometric measurements, and bioelectrical impedance analysis (BIA) (115); similar results were described by Laing et al (35415851) and Borre et al (29373941) in their cross-sectional studies, who found 29% and 25% of patients with NET malnourished, respectively. A lower prevalence of malnutrition was assessed by Qureshi et al, who found this condition in 14% of outpatients with GEP-NET (27147385). However, the authors recommended a routine use of malnutrition screening in all patients with GEP-NET.”

Page 9, line 386-7 and line 393-395: Please add that the role of an accredited Dietitian is important to provide nutrition assessment and management.

  1. Thank you. We added that the role of an accredited Dietitian is important to provide nutrition assessment and management.

Page 9, line 390-392: Please expand on how hormone production affects nutritional status of NETs, it is due to symptoms arising from hormonal syndromes.

  1. Thank you. We added the following sentence: “In this regard, the clinical manifestations of NETs could be characterized by various hormone hypersecretion syndromes such as the carcinoid syndrome (CS) or the hyperin-sulinaemic hypoglycaemia syndrome in case of insulinomas (29020456).”

Page 9, section 3.1.1 and 3.1.2: the introductory information included here is repetitive with previous sections so can reduce

  1. Thanks for your suggestion. We shortened the introduction of these sections.

Page 10, line 412: More emphasis needs to be placed on the strength of evidence here. I'm not sure you can say that a MD may prevent cancer agressivness after results of one study?

  1. We now added this sentence: “Obviously, the strength of evidence is related only to one study and further studies are needed to confirm this finding.”

Page 10, line 414-418: Please adjust wording so it is clear these recommendations are for cancer prevention and not for patients with current disease

  1. Thank you. We specified that the recommendations have been proposed “in patients without current disease”.

Page 10, line 434: Reference 126 cited with "this effect may be particularly useful in some form of secreting NETs...." is not relevant to glucagonomas (the article is regarding glucagon in general) and should be taken out along with the text.

  1. We removed “such as glucagonoma”.

Page 10, section 3.1.2: The authors should note that a high fat diet can worsen or induce symptoms in NET patients, as often they are suffering from fat malabsorption, so please re-word this section with this consideration.

  1. Thank you. We added this consideration: “On the other hand, it should be also considered that a high fat diet can worsen or induce symptoms related to fat malabsorption; therefore, caution is needed in KD use.”

Page 10, section 3.1.3: Please re-word any statements that indicate that an intermittent fasting diet may be beneficial or recommended for NET patients, as there is not enough evidence to make this recommendation.

  1. Thank you. We rewrite the paragraph as follows: “However, based on studies in non-cancer patients, we can only speculate that the anti-inflammatory effects and overweight and obesity reduction related to IF could improve the tumor responses to anticancer treatment also in NET patients (130). Furthermore, it has been supposed that IF could induce an anti-Warburg effect similarly to KD (128). Obviously, this is an assumption without enough evidence to make recommendation, and further studies are needed.”

Page 10, line 453: Please change to "due to treatment-induced pancreatic insufficiency"

  1. We modified in “due to treatment-induced pancreatic insufficiency”, as suggested.

Page 10, line 453: "It would be advisable to avoid..." this statement refers to a study that was not done in NET patients, so please ensure this is clear.

  1. We modified the sentence as follows: “It would be advisable, extending the evidence deriving from non-NET patients, to avoid sources of histamine”.

Page 10, line 455 onwards: Much of the evidence for avoiding amine foods is anecdotal, please ensure this is stated

  1. We now stated that “these indications still require strong scientific evidence.”

Page 10, line 461: "assumption" change to "consumption"

  1. We modified "assumption" to "consumption"

Page 11, line 482: malnutrition and cachexia are different with different definitions, please revise so it's clear

  1. Thank you for your observation. We modified the sentence as follows: “Malnutrition, which is a deficiency or imbalances in a person's intake of energy and/or nutrients, and cachexia, which is characterized by a loss of fat and muscle mass, frequently occur in cancer patients”

Page 11, section 3.2.5: Should this read 'NET treatment-related symptoms'?

  1. Thanks for your suggestion. We modified the title in “'NET treatment-related symptoms'”

Reviewer 2 Report

The manuscript subject is very valid and relevant. Nutritional choices have a significant impact on NET management and patients' quality of life, thus this aspect of NET care should be further investigated to provide more robust evidence. Nutritional guidelines should be an integral part of the multidisciplinary approach in NET management.   

Author Response

On behalf of all the authors, I would like to kindly thank the Reviewer for the positive comments on the manuscript. Thank you very much and our best regards.

Sincerely yours,

Dr Giovanna Muscogiuri

Round 2

Reviewer 1 Report

Thank you to the authors for their thorough review and response to my comments and recommended changes. The modifications made to the manuscript are appropriate and strengthen the overall paper.